# An Opposition-Based Learning Black Hole Algorithm for Localization of Mobile Sensor Network

**DOI:** 10.3390/s23094520

**Published:** 2023-05-06

**Authors:** Wei-Min Zheng, Shi-Lei Xu, Jeng-Shyang Pan, Qing-Wei Chai, Pei Hu

**Affiliations:** College of Computer Science and Engineering, Shandong University of Science and Technology, Qingdao 266590, China

**Keywords:** opposition-based learning, black hole algorithm, mobile node localization, wireless sensor network

## Abstract

The mobile node location method can find unknown nodes in real time and capture the movement trajectory of unknown nodes in time, which has attracted more and more attention from researchers. Due to their advantages of simplicity and efficiency, intelligent optimization algorithms are receiving increasing attention. Compared with other algorithms, the black hole algorithm has fewer parameters and a simple structure, which is more suitable for node location in wireless sensor networks. To address the problems of weak merit-seeking ability and slow convergence of the black hole algorithm, this paper proposed an opposition-based learning black hole (OBH) algorithm and utilized it to improve the accuracy of the mobile wireless sensor network (MWSN) localization. To verify the performance of the proposed algorithm, this paper tests it on the CEC2013 test function set. The results indicate that among the several algorithms tested, the OBH algorithm performed the best. In this paper, several optimization algorithms are applied to the Monte Carlo localization algorithm, and the experimental results show that the OBH algorithm can achieve the best optimization effect in advance.

## 1. Introduction

The wireless sensor network (WSN) is a distributed sensor network in which the number of nodes can reach hundreds or thousands [1]. Different network topologies are formed between nodes, and nodes can communicate with each other. The node is composed of anchor nodes and common nodes [2]. The common node senses the external environment through its hardware equipment, collects data, and transmits it to the anchor node. The anchor node filters and fuses the data and sends the sorted data to the network owner through the network. The network owner responds to the changes in the environment based on the collected data. However, if the anchor node does not know the location of the node sending the information, the received information is meaningless. Therefore, location is not only a critical issue of wireless sensor networks but also the basis for the subsequent operation of the network.

There are many algorithms to realize an unknown node location in WSN, such as the DV-Hop algorithm, the APIT algorithm, and the centroid algorithm. The DV-Hop algorithm was proposed by Niculescu et al. [3]. The main idea is to use the minimum hop count and average hop distance between the unknown node and the anchor node to replace the distance between the two. When the distance between the unknown node and the other three anchor nodes is obtained, you can use the maximum likelihood method or the trilateration method to solve the position of the unknown node. The APIT algorithm was proposed by Tian He et al. [4]. The main idea is to continuously narrow the range of the area where the unknown node is located according to whether the unknown node is in the triangle area formed by three adjacent anchor nodes and to take the centroid position of the finally locked area as the coordinates of unknown node. The centroid algorithm was proposed by Nirupama et al. [5]. This algorithm calculates the position of unknown nodes through the connectivity of the network. For an unknown node, the algorithm uses the anchor nodes around the unknown node as the polygon vertices of the unknown node and calculates the centroid position of the polygon, and the obtained result is the coordinate of the unknown node.

With the gradual maturity of mobile terminal and mobile internet technology, the localization of mobile wireless sensor networks (MWSN) has become a new trend of current wireless sensor networks and a hot research field of WSN [6]. The most significant difference between MWSN and WSN is that the nodes constantly move. The mobility of nodes can be used to increase the network coverage, improve the network scalability and reliability, and also put forward new requirements for the location algorithm. Because the network’s topology is constantly changing, MWSN requires that nodes can be dynamically located. Otherwise, the location information of nodes will become invalid with time, and MWSN will not be able to operate normally [7]. Many mobile node localization algorithms have been proposed, such as particle filter algorithm and Monte Carlo localization (MCL) algorithm [8].

The swarm intelligence algorithm [9] simulates the behavior of living things in nature [10]. It is a group of algorithms with self-organization abilities and self-learning abilities and has the characteristics of adaptability and parallelism. This algorithm has fewer requirements for optimization problems, is simple to use, and is suitable for solving large-scale problems. In 1989, Gerardo Beni and Jing Wang [11] of the University of California proposed the concept of "swarm intelligence". The basic principle is to simulate the behavior of animal groups in nature and to take advantage of the cooperation and communication between animal groups to achieve the optimization goal. Unlike algorithms with complex internal designs, swarm intelligence algorithms are simple and have stronger robustness and adaptability. Therefore, once the concept of swarm intelligence was proposed, it attracted widespread attention. Standard swarm intelligence algorithms include brain storm optimization (BSO) [12], particle swarm optimization (PSO) [13], the firefly algorithm (FA) [14], the whale optimization algorithm (WOA) [15], grey wolf optimization (GWO) [16], and the black hole (BH) algorithm [17], etc.

Metaheuristic search algorithms [18] are divided into the following four main categories: evolutionary algorithms, swarm intelligence algorithms, human-based algorithms, and physics-based algorithms. Evolutionary algorithms are a class of optimization algorithms based on the principles of biological evolution in nature, which are used to find optimal or near-optimal solutions in the search space. Evolutionary algorithms mainly include evolutionary programming (EP) [19], the genetic algorithm (GA) [20,21], genetic programming (GP) [22], etc. Swarm intelligence algorithms are a class of optimization algorithms based on the behavior of groups in nature, which achieve global search or optimization problem solving by simulating the collaboration and cooperation among individuals in a group. Swarm intelligence algorithms include particle swarm optimization (PSO) [13,23], grey wolf optimization (GWO) [16,24,25], the whale optimization algorithm (WOA) [15], etc. Human-based algorithms are a class of optimization algorithms based on human behavior and cognition, which apply human intelligence and experience to the process of problem-solving. Human-based algorithms include the Jaya algorithm (JA) [26], human-inspired algorithm (HIA) [27], teaching-learning based optimization (TLBO) [28], etc. Human-based algorithms can fully use human intelligence and experience in practical applications and are particularly advantageous in complex, uncertain, and multi-objective problems. Physics-based algorithms are a class of optimization algorithms that mimic physical phenomena and principles of nature for problem-solving and optimization. These algorithms usually achieve problem-solving or optimization by simulating physical processes, mechanical laws, energy transfer, etc. These algorithms can achieve global optimization by simulating the physical phenomena and laws of nature and have strong robustness and global search capability in some cases. Physics-based algorithms mainly include the gravitational search algorithm (GSA) [29], the multi-verse optimizer (MVO) [30], and the black hole (BH) algorithm [17], etc.

Mobile node localization problems usually involve searching for the location of object nodes in complex spaces, such as locating the location of moving objects in three-dimensional space. The search space can be very large and complex, and traditional exact search methods are often not suitable. The metaheuristic search algorithm can efficiently search in the complex search space by simulating the optimization process in nature. Mobile node localization problems are usually performed in dynamic environments, where the location of the target node may change with time and other factors. Metaheuristic search algorithms are usually adaptive and can adjust search strategies in real time in dynamic environments to adapt to changes in target locations. Metaheuristic search algorithms are generally robust and adaptable and can cope with various uncertainties and noises. In the mobile node localization problem due to the uncertainty of the environment and nodes, such as noise, sensor error, communication interruption, etc., metaheuristic search algorithms can usually find better solutions in these uncertainties.

Heuristic algorithms usually search in the search space through the population to replace the solution process of the problem to be optimized. Individuals in the population represent candidate solutions to the problem to be optimized, and the performance of candidate solutions corresponds to fitness. Almost all heuristic algorithms simulate the behavior patterns of creatures in nature or this natural phenomenon. The main idea of the black hole algorithm is to find the optimal solution of the problem to be optimized in the search space by simulating the black hole to attract the moving side of stars.

In this paper, we combine the BH algorithm and MCL algorithm to localize mobile nodes in 3D WSN. The principle of the BH algorithm is simple, and there are few parameters. Using it to optimize the positioning of wireless sensor networks will not cause a large burden on its memory. It is precisely because the BH algorithm only simulates the phenomenon that the black hole attracts stars to move and does not balance the exploration and development stages of the algorithm, resulting in weak optimization ability and slow convergence speed of the algorithm. In order to overcome the defect of BH algorithm, we propose an opposition-based learning black hole (OBH) algorithm. According to the evolution degree of the population, OBH divides the population’s evolution into four stages. It applies different opposite strategies to the population at different evolution stages. At the same time, this paper uses an adaptive strategy [31] to determine the evolution interval of the algorithm so that the population adopts a more accurate opposition-based learning strategy (OBL).

This section describes the organizational structure of the article. Section 2 introduces the basic principles of the BH and MCL. Section 4 introduces how to combine the reverse learning strategy with the BH algorithm. In Section 5, this paper tests the OBH’s performance and compares it with several swarm intelligence algorithms. Section 6 is the experimental simulation part. Finally, the full text is summarized in Section 7.

## 2. Related Work

### Monte Carlo Localization Algorithm

The MCL [32] algorithm was initially used in mobile robot localization, which is divided into two stages: prediction and filtering. The key idea is to use a group of weighted samples to represent the posterior distribution of the possible positions of robots. In the prediction stage, the algorithm mainly relies on the position information of the robot in the previous stage to predict the possible position of the robot after moving. However, the uncertainty of the prediction position gradually increases when the robot continuously moves, which makes the prediction of the robot’s position difficult. In the filtering stage, the robot may observe landmarks with known positions during its movement, so the position information of these landmarks can filter the wrong position information obtained in the prediction stage.

The localization of sensor nodes in MWSN differs significantly from that of mobile robots. First, the active terrain of the robot is known, while the sensor nodes work in the unknown terrain. Secondly, the robot can control its motion posture very well. The speed and direction of motion are known. However, the sensor nodes rarely or almost never control their movement, so it is difficult to obtain the speed and direction of movement. The third difference is that the landmark information observed by the robot in the process of moving is accurate, which can effectively assist the current robot in locating the position. The sensor node only knows that the landmark information observed is within its radio range, and the specific location information cannot be known. Finally, the computing power of the robot is much higher than that of the sensor node. Because of the delay control of the sensor network, the sensor node cannot perform large-scale complex computing.

Although the difference between the computing power and ranging accuracy of sensor nodes and robot localization makes the positioning of sensor nodes more complex, the network scale of wireless sensor network mobile node positioning is more significant. Many nodes can share each other’s location information, which is beneficial to the mobile positioning of sensor nodes.

In the localization of MWSN, the node leaves the previous location to reach the current location. For the convenience of prediction, this paper assumes that the time is discrete, and the node needs to be located in each time unit. Because the direction and speed of node’s movement are unknown, the previous location information becomes more and more inaccurate after the node moves. On the other hand, the observation value from the anchor node in the sensor network can help us filter some false predicted locations. The samples are independent of each other, and the weight of the samples is 0 or 1. Algorithm 1 shows the pseudocode of MCL.
**Algorithm 1** Monte Carlo localization algorithm1:**Initialization:** Initialize the wireless sensor network. N is the node’s number, $L_{0}$ = {set of random locations in the deployment area}2:**Step:** According to the node position $L_{t-1}$ obtained from the previous time unit and the current observation value $o_{t}$, calculate the node position $L_{t}$ in the current time unit3:**while** (size($L_{t}$) < *N*) **do**
(1)$$R=\left\{l_{t}^{i}|l_{t}^{i}\right.\mathrm{is}\;\mathrm{selected}\;\mathrm{from}\;p\left(l_{t}|l_{t-1}^{i}\right),l_{t-1}^{i}\in L_{t-1},\left.1⩽i⩽N\right\}.$$
(2)$$R_{\mathrm{fitered}\;}=\left\{l_{t}^{i}|l_{t}^{i}\right.\mathrm{where}\;l_{t}^{i}\in R\;\mathrm{and}\;\left.p\left(o_{t}|l_{t}^{i}\right)>0\right\}.$$
(3)$$L_{t}=\mathrm{choose}\left(L_{t}\cup R_{\mathrm{fittered}\;},N\right).$$4:**end while**

$l_{t}$ is the possible location set of unknown nodes predicted at time *t*; $o_{t}$ represents the observation of anchor nodes at time *t*; and $p\left(l_{t}|l_{t-1}\right)$ represents the probability of predicting the location of the node at time t−1 according to the location of the unknown node at time t−1, with the formula shown in the Equation (Equation 4). $p(l_{t}|o_{t})$ represents the probability that the current node is at $l_{t}$ according to the observation. Therefore, we could take advantage of $l_{t}$ to predict the location distribution of nodes at time *t* and then filter the impossible location according to $o_{t}$. Finally, the possible position distribution of *N* samples $L_{t}$ is obtained. The filter distribution on each time unit needs to be calculated. The set of *N* samples $L_{t}$ represents the final position distribution of the node. The MCL calculates recursively in each time unit, and $L_{t-1}$ represents all previous observations. Therefore, the results can be obtained only using $L_{t-1}$ and $o_{t}$.
(4)$$p\left(l_{t}|l_{t-1}\right)=\left\{\begin{array}{cc} \frac{1}{\pi v_{\max}^{2}} & \mathrm{if}\;d\left(l_{t},l_{t-1}\right)<V_{\max}\\ 0 & \mathrm{if}\;d\left(l_{t},l_{t-1}\right)\geq V_{\max} \end{array}\right..$$

As the anchor node moves in Figure 1, the unknown node in region *A* will not be detected by it, the unknown node in region *B* will enter the detection range of the anchor node, and the unknown node in region *C* will always be in the detection range of the anchor node. The node adjacent to the current anchor node can also convey similar information about the approximate location of the unknown node.

If unknown nodes can be detected by anchor nodes, the distance between them is within *r*. If unknown nodes are not directly monitored by the anchor node but can be monitored by the neighbor of the anchor node, the distance between them is within [r,2r].

Therefore, if the unknown node *l* is not in the monitoring region of the anchor nodes *T* but in the detection region of the neighbor nodes *N*, anchor node *S* can directly monitor *l*, the location of the unknown node can be filtered according to the following formula:(5)filter(l)=d(l,S)≤r∧d(l,N)≤r∧r<d(l,T)≤2r.

## 3. Background

### Black Bole Algorithm

Black holes are celestial bodies that exist in nature and constantly attract objects around them. In terms of the horizon of the black hole, also known as the radius of the black hole, once an object enters it, it will never escape. The BH algorithm is based on this characteristic of black holes. The individuals in the BH [33] algorithm are called stars, and the optimal solution is called a black hole. The radius of the black hole is calculated as follows:(6)$$R=\frac{f_{BH}}{\sum_{i=1}^{N}f_{i}}.$$

Here, $f_{BH}$ is the fitness of the black hole, $f_{i}$ is the fitness of the i-th star, and *N* is the population size. When the black hole attracts the stars around it, the position update formula of the stars is as follows:(7)$$p_{i}\left(t+1\right)=p_{i}\left(t\right)+rand\times \left(p_{BH}-p_{i}\left(t\right)\right)\;i=1,2,\ldots ,N.$$

$p_{i}\left(t\right)$ is the position of the i-th star at iteration *t*, $p_{BH}$ is the position of the black hole in the population at iteration *t*, and rand is a random number between 0 and 1.

The black hole will devour the stars entering its radius. In order to keep the number of populations unchanged, a new star will be randomly generated. If no new individuals are regenerated after the black hole devours the star, the population will drop sharply in the later stage of the algorithm because the black hole continues to devour the star. At this time, a small number of individuals cannot undertake the task of exploring and developing the search space, which will seriously affect the final performance of the algorithm.

A pseudocode of the BH algorithm is shown in Algorithm 2, which shows that the BH algorithm has a simple process and few parameters, which is more suitable for wireless sensor network localization applications.
**Algorithm 2** Black Hole algorithm1:**Initialization:** Generate individuals randomly2:**while** (t < MaxGeneration) **do**3:    Calculate the fitness of the population on the test function set4:    Select black hole individuals according to fitness5:    Move individuals in the population according to the Equation (Equation 6)6:    If the fitness of individual after moving is better than that of the black hole, it will become a new generation of black holes7:    The star entering the radius of the black hole is swallowed by it, and the algorithm will randomly generate a new star.8:    t = t + 19:**end while**

## 4. Opposition-Based Learning Black Hole Algorithm

The OBL strategy [34] is to improve the learning rate. The opposite operation can fully explore the search space, quickly find a promising region, and accelerate the population toward global optimization.

Define *x* as an individual in the population. Assume that the upper and lower bounds of *x* are *a* and *b*, respectively. The reflective opposition of *x* is as follows:(8)$$x_{o}=a+b-x.$$

The relationship between *x*, $x_{o}$, *a*, and *b* is shown in the Figure 2.

Divide the interval in Figure 2 to obtain the other three kinds of opposition about *x*, namely, $x_{qo}$ (quasi-opposition of *x*), $x_{so}$ (super-opposition of *x*), and $x_{qr}$ (quasi-reflective opposition of *x*), which are calculated by the following formula:(9)$$x_{qo}=rand\left(c,x_{o}\right),$$
(10)$$x_{so}=\left\{\begin{array}{c} rand\left(x_{o},b\right),x<c\\ rand\left(a,x_{o}\right),x\geq c \end{array}\right.,$$
(11)$$x_{qr}=rand\left(x,c\right).$$

Figure 3 clearly shows the four opposition forms of *x* on the interval [a,b]:

Further observation of Figure 3 shows that no opposition rule is defined on the interval [a,x]. Through experiments, this paper finds that it is necessary to further explore the interval [a,x]. There will also be promising points on this interval, which can considerably improve individual performance and help break away from local optimization. Therefore, we propose a new opposition rule called near opposition ($x_{no}$), which is defined on the interval [a,x].

In the opposition strategy, parameter Jr represents the probability of the individual being the opposite. If the value of Jr is too large, the opposite individuals will be generated frequently, which makes the population repeat the opposite operation in the search space, wasting time in evaluating the fitness function. If Jr is too small, the number of opposite individuals generated is too small to explore the unknown area fully. In this paper, the initial value of Jr is 0.8, eventually reaching 0.4 as the iteration gradually decreases.

The evolution process of the population can be divided into four states, namely, exploration [35], exploitation, convergence, and jumping out. Exploration and development are essential parts of the swarm intelligence algorithm. It is an essential condition for the algorithm to realize intelligence. Exploration refers to the group searching for promising areas in the search space. In contrast, exploitation [36] refers to the full development of promising areas to find the best performance position in the area. Too much exploration will increase the randomness of the algorithm, while too much exploitation will reduce the randomness of the algorithm. Exploration and exploitation are critical to the algorithm, so to make the algorithm achieve better performance, we need to balance the two. Convergence means that the current population has found a better global optimal value. At this time, all individuals in the population are close to the new global optimal value. Jumping out means that the current population has fallen into the local optimal value but found a better performance position in the latest evolution. However, this position is relatively far from the local optimal position. At this time, all individuals in the population are close to the better-performance individuals, which is jumping out of state.

Since the population is divided into four states, determining the state of the population becomes very important. Obviously, the average distance between individuals in a population is a good criterion for judging the state. The average distance between all individuals and the i-th individual is calculated using the following formula:(12)$$d_{i}=\frac{1}{N-1}\sum_{j=1,j!=i}^{N}\sqrt{\sum_{k=1}^{D}{\left(x_{i}^{k}-x_{j}^{k}\right)}^{2}}.$$

*D* represents the dimension that the individual has, and *K* represents a certain dimension of the individual. The average distance between the globally optimal individual and the other individuals is the smallest under the convergence state because individuals in the population only surround the global optimal individual. The average distance is the largest when jumping out because the global optimal individual may be far away from other individuals in the population. This paper lets $d_{g}$ represent the average distance between all individuals and the globally optimal individuals. The maximum value $d_{max}$ represents the average distance between individuals and others in the group, and $d_{min}$ represents the minimum value.
(13)$$\mu_{S_{a}}\left(f\right)=\left\{\begin{array}{cc} 0, & 0\leq f\leq 0.4\\ 5\times f-2, & 0.4<f\leq 0.6\\ 1, & 0.6<f\leq 0.7\\ -10\times f+8, & 0.7<f\leq 0.8\\ 0, & 0.8<f\leq 1. \end{array}\right.$$
(14)$$\mu_{S_{b}}\left(f\right)=\left\{\begin{array}{cc} 0, & 0\leq f\leq 0.2\\ 10\times f-2, & 0.2<f\leq 0.3\\ 1, & 0.3<f\leq 0.4\\ -5\times f+3, & 0.4<f\leq 0.6\\ 0, & 0.6<f\leq 1 \end{array}\right.$$
(15)$$\mu_{S_{c}}\left(f\right)=\left\{\begin{array}{cc} 1, & 0\leq f\leq 0.1\\ -5\times f+1.5, & 0.1<f\leq 0.3\\ 0, & 0.3<f\leq 1 \end{array}\right.$$
(16)$$\mu_{S_{d}}\left(f\right)=\left\{\begin{array}{cc} 0, & 0\leq f\leq 0.7\\ 5\times f-3.5, & 0.7<f\leq 0.9\\ 1, & 0.9<f\leq 1 \end{array}\right.$$

Equations (Equation 13)–(Equation 16) describe the fuzzy membership functions of exploration, development, convergence, and jump-out stages, respectively. The following formula can show the evolution factor *f* judging the population state:(17)$$f=\frac{d_{g}-d_{\min}}{d_{\max}-d_{\min}}\in \left[0,1\right].$$

During the iteration, the population is in the exploration stage, and *f* is at a significant value. As the iteration progresses, the population enters the exploitation stage, and *f* starts to decrease gradually until the population enters the convergence state, and *f* also reaches the minimum value. Once the algorithm finds a better individual in another location, the population enters the jumping state, and *f* increases sharply until the population enters the convergence state again. Many test results have proved a specific range of intersection between each interval. Figure 4 clearly describes how the evolution factor *f* describes the state of the population.

The transition rule between states is S1⇒S2⇒S3⇒S4⇒S1. Assuming that *f* is currently in the crossing interval of S1 and S2 and the previous state is S4, it can be seen that the current population has entered the exploration state. After knowing the evolution state of the current population, we can take corresponding opposite strategies for the population at the appropriate time. This method of obtaining the population state according to the population evolution factor *f* and then adopting different strategies for the population is also called an adaptive strategy.

When the algorithm encounters a deceptive location, the global optimal individual will always hover at that location, making the algorithm always explore a worthless area, wasting much fitness evaluation times. In order to make the OBH algorithm take advantage of the optimization opportunities as much as possible and avoid the endless exploration and development in the worthless areas, this paper will use Levy flight to accelerate the particles to jump out of the local optimal. Levy flight is a unique random walk, meaning an individual can advance any distance. Figure 5 shows the step size of the Levy flight, each line segment represents a random walk of the individual.

It can be seen that the Levy flight has a significant probability of being of a large step size. When an individual falls into a local optimum, the Levy flight can be used to let the individual jump out of the current position and find a new optimal position through a large step size. Moreover, the Levy flight not only helps individuals jump out of the local optimum but also allows individuals to move forward at a significant pace in the exploration stage and to find promising areas in the whole search space as soon as possible after exploration.

In order to avoid the profound negative impact on the population caused by the elite individuals trapped in the local optimal, this paper introduces several elite individuals to the OBH algorithm. At the same time, we found that the BH algorithm did not fully use the achievements of previous generations of individuals in the search space, so we introduced the inertia weight ω in the algorithm to use the current population information fully. The experiments show that the two improved methods can improve the algorithm’s performance. The following is the position update formula of individuals in the OBH algorithm:(18)p(t+1)=p(t)+ω×rand×(BH1−p(t)+BH2−p(t)+BH3−p(t)).

The flow of the OBH algorithm is described in Algorithm 3. At each iteration, the population is moved according to the evolutionary logic of the black hole algorithm. Then, the population will be sorted according to fitness. In order to retain more useful information in the population, this paper retains the best *k* elite individuals in the population. In order to perform reverse operations on the remaining individuals, it is necessary to calculate the evolutionary state of the current population. Finally, according to Jr and population evolution status, corresponding opposite operations are generated for individuals and their fitness is evaluated.
**Algorithm 3** OBH algorithm.1:**Initialization:** popsize, Dimension *D*, max_iter, Jump Rate Jr, The optimal number of individuals to be reserved *k*2:**while** (t < MaxGeneration) **do**3:    **for** i=1; i<=popsize; i++ **do**4:        Calculate the fitness of the population on the test function set5:        Select black hole individuals according to fitness6:        Update the population location according to the Equation (Equation 18)7:        If the performance of an individual after moving is better than that of a black hole, it will become a new generation of black holes8:    **end for**9:    Sort the population after one iteration and keep the top *k* individuals10:    Calculate the average distance according to the Equation (Equation 12)11:    Calculate population evolution factor *f* according to the Equation (Equation 17)12:    Judge the state of population according to the previous state of the population and *f*13:    **for** i=k+1; i<=popsize; i++ **do**14:        Generate a random number *p*15:        **if** p≤Jr **then**16:           Generate corresponding opposite individuals according to the current state of the population17:           Evaluate the individuals after opposition. If the performance is better than the global optimal solution, it will become the new generation of black hole18:        **end if**19:    **end for**20:    t = t + 121:**end while**

Algorithm 4 describes how to select the appropriate opposite rule for individuals according to the current evolutionary state of the population. It is worth noting that the probability of imposing reverse rules on individuals is not fixed and needs to be constantly adjusted. At the same time, individuals with low fitness gain more benefits from severe reverse operations than individuals with high fitness.
**Algorithm 4** Reverse rules.1:**Initialization:** population evolution factor *f*, individual *x*2:When the population is in the exploration state, it reduces the probability of $x_{no}$ of the population while increasing the probability of $x_{so}$ of the population.3:When the population is in a converged state, it reduces the probability of $x_{so}$ of the population while increasing the probability of $x_{no}$ of the population.4:When the population is in the development state, the probability of $x_{no}$ and $x_{qr}$ of the population can be appropriately increased, otherwise the probability of $x_{qo}$ and $x_{qr}$ of the population can be appropriately increased.
(19)$$fitness\left(i\right)=\sum_{i=1}^{N}\sqrt{\sum_{d=1}^{D}{\left(RPos(d)-VPos(d)\right)}^{2}}.$$

Equation (Equation 19) describes the fitness function when OBH optimizes MCL, where *D* is the dimension of the individual and *N* is the number of unknown nodes. The OBH algorithm is used to optimize the approximate position VPos of the unknown node obtained by MCL positioning and to finally obtain a more accurate position RPos of the unknown node. Therefore, it is necessary to take VPos as the input and output the optimized precise position RPos as the result. Since this is a simulation experiment, the real positions of unknown nodes are known. When OBH optimizes MCL, the fitness function is the sum of the distances between the approximate positions of all nodes and the real positions.

For swarm intelligence algorithms, the number of fitness evaluations is a precious resource. In the OBH algorithm, each individual may perform multiple fitness evaluations during each iteration. The opposite individuals generated must also be evaluated. Therefore, in order to ensure that the total number of fitness evaluations remains unchanged, we need to reduce the number of iterations of the algorithm.

Fitness evaluation is the most time-consuming operation in the algorithm, so when calculating the time complexity of the algorithm, the number of fitness evaluations should be used as a measure. The time complexity of calling the BH algorithm each time is O(N∗T), where *N* is the size of the population and *T* is the number of iterations each time. The Levy flight is added to the OBH algorithm. Since the Levy flight will not increase the number of fitness evaluations of the algorithm, it will not cause an increase in the complexity of the algorithm. When judging the evolutionary state of the population, there is a N∗N∗D cycle in the algorithm, where *D* is the dimension of the individual, but judging the evolutionary state of the population will not perform fitness evaluation, so it will not significantly increase the running time of the algorithm. When the opposite operation is performed on the individual, the fitness function will be evaluated. In order to ensure that the number of fitness evaluations of the OBH algorithm is consistent with that of the BH algorithm, the total number of iterations of the OBH algorithm will be reduced accordingly. Therefore, the time complexity of the OBH algorithm is also O(C∗N∗T), and *C* is a number less than 2. According to the above discussion, compared with the performance improvement brought by the OBH algorithm, the increased running time of the OBH algorithm is acceptable.

## 5. Experimental Analysis of Algorithm

This section first illustrates the improvement in the OBL strategy to the BH algorithm by comparing OBH with BH. Then, OBH is compared with other first-class swarm intelligence algorithms to highlight the outstanding performance of the OBH algorithm. At the end of this section, this paper analyzes the algorithm’s convergence through the fitness function curve. All algorithms in this section are tested on the CEC2013 test function set.

### 5.1. Comparison between OBH Algorithm and BH Algorithm

This section first uses Friedman’s test [37] to verify the improvement in the OBL strategy on the BH algorithm. Friedman’s test can give a score to the algorithm. The algorithm with excellent performance has the lowest score. The experimental results are shown in the following table:

As seen from Table 1, in terms of score, the performance of the OBH algorithm is much better than BH. Compared with BH, the OBH performance is improved by 49%.

Next, the results of the OBH and BH algorithms are measured at a significant level at α=0.05 under Wilcoxon’s signed rank test. When BH performs better than OBH, it is represented by >, < represents the opposite meaning, and = represents the same performance for the two algorithms.

We can see from Table 2 that the BH algorithm only slightly outperforms the OBH algorithm on the function $f_{26}$. In the remaining 27 test functions, the performance of the OBH algorithm is far better than that of the traditional BH algorithm, which shows that the OBL strategy significantly improves the performance of the BH algorithm. The simple logic of the BH algorithm makes it very suitable for application in various fields. If the algorithm wants to effectively explore the space, exploit promising areas in the search space, quickly move towards the current optimal position, and quickly jump out of the local optimal, it is not feasible to rely on the simple logic of the BH algorithm alone. The OBL strategy added to the algorithm can improve the learning rate of the BH algorithm. In short, under the same number of iterations, the OBH algorithm can use the experience gained more effectively and quickly to find a better solution. Even the deviation in the process of optimization does not matter. Depending on the error correction mechanism designed in this paper, the OBH algorithm can return to the correct direction in time and find the global optimal position.

### 5.2. Comparison with Common Optimization Algorithms

To make the improvement in the algorithm more convincing, this paper chooses to compare the OBH algorithm with five outstanding swarm intelligence optimization algorithms, such as the bat algorithm (BA) [38,39], the beluga whale optimization (BWO) [40], fish migration optimization (FMO) [41], snake optimization (SO) [42], and comprehensive learning particle swarm optimization (CLPSO) [43]. The test data are presented in Table 3.

On the 28 test functions of the CEC2013, the OBH algorithm outperformed BA on 21 functions and BWO on 27 test functions, and CLPSO outperformed OBH on only 10 test functions, but on the remaining 14 functions, CLPSO performed worse than the OBH algorithm. Compared with the FMO algorithm, the OBH algorithm performs better on 18 functions. The OBH algorithm performs better than SO on 17 functions and is inferior to SO on only ten.

The OBH algorithm performs better than many excellent swarm intelligence algorithms, and the advantages of the OBL strategy will be further described in combination with the fitness curve.

We selects nine functions in the test function set to demonstrate the convergence of the algorithm in Figure 6. The performance of the other five algorithms is generally inferior to OBH. Even if they can achieve the same effect as the OBH algorithm in a specific function, the OBH algorithm can also find the optimal value faster than they can. We can see from Figure 6 that the fitness curve of the OBH will decline rapidly at a certain time, which shows that the jumping out of the local optimization mechanism designed for the OBH algorithm in this paper highlights its power and benefits from the adaptive mechanism, which can accurately judge the evolution state of the algorithm and select the appropriate reverse strategy for the algorithm so that the algorithm can find the global optimization as soon as possible.

## 6. Application in Monte Carlo localization

This section applies the OBH algorithm to MCL and replaces the general algorithm in Section 5.2 with several excellent algorithms to exploit the potential of the OBH algorithm further, such as adaptive particle swarm optimization (APSO) [44], differential evolution (DE) [45], GWO, and PSO. In each group of experiments, the number of nodes remained unchanged, with a total of 200, and the number of anchor nodes gradually increased from 5 to 30.

Figure 7 shows the comparison results of the OBH algorithm and the other six intelligent algorithms applied in MCL in a deployment area of 200 m × 200 m × 200 m. The y-axis represents the mean value of positioning error. As the number of anchor nodes increases, each population intelligence algorithm’s positioning accuracy significantly improves. The average positioning accuracy of seven algorithms optimizing MCL are shown in Table 4. The localization accuracy of the OBH algorithm is always the highest, which shows that the positioning accuracy of this algorithm is the best of the seven algorithms.

To further verify the performance of the OBH algorithm, the deployment area of the 3D experiment is further expanded to 400 m × 400 m × 400 m. Figure 8 shows the comparison of the positioning accuracy of seven algorithms:

As seen from Figure 8, OBH is also the best among the seven algorithms in a larger deployment area. Even if the number of anchor nodes in the deployment area is only five, OBH can still effectively eliminate the interference of external error information, give full play to the power of the algorithm, and further improve positioning accuracy. With the increased anchor node density, the OBH algorithm still maintains considerable advantages. It can optimize the positioning results of the MCL algorithm to the greatest extent and achieve the best results in the seven population intelligence algorithms.

The average positioning accuracy of seven algorithms optimizing MCL are shown in Table 5. It can be seen that in the deployment area of 400 m × 400 m × 400 m, the best positioning accuracy can be obtained by using OBH to optimize the unknown node position obtained by MCL.

When the increase in the deployment area leads to a decrease in the density of anchor nodes, the available information transmitted by the anchor nodes in the network is very small, and OBH can use this small amount of useful information to be optimized in the search space. It can be seen that the approximate position of the unknown node obtained by MCL positioning is passed to seven optimization algorithms for optimization, and OBH can perform the best solution in the search space.

## 7. Conclusions

The BH algorithm has a simple structure and fewer parameters, making it more suitable for wireless sensor network localization with limited memory and computing power. However, the traditional BH algorithm performs generally and cannot stand out among many swarm intelligence algorithms. In this paper, we combine the OBL strategy and BH algorithm, and the OBH algorithm is proposed.

The OBH algorithm reduces the number of duplicate individuals as much as possible based on preserving the dominant individuals, fully uses the current population information, and avoids wasting valuable fitness evaluation time. At the same time, an adaptive strategy can accurately judge the evolution state of the population, formulate the best OBL strategy for the algorithm, and significantly improve the performance of the algorithm. To verify the improvement in the OBH algorithm, this paper compares five excellent swarm intelligence algorithms with OBH, and the results show that the OBH algorithm performs best among the six algorithms. Finally, this paper applies the OBH and the other six algorithms to MCL. The simulation results show that the MCL positioning accuracy optimized by OBH is the highest, which indicates that the performance of the OBH algorithm is first class.

Compared with the BH algorithm, the performance improvement in the OBH algorithm is very obvious, but it comes at a price. At each iteration, the algorithm calculates the evolutionary state of the population and operates oppositely, which increases the runtime of the algorithm. In a certain period of time, there are populations and opposite populations in the algorithm at the same time, which also increases the demand for memory resources from the sensor nodes to a certain extent.

It is inevitable that the opposite operation will increase the runtime of the algorithm, but we can reduce the memory resource requirements of the algorithm through the compact strategy. The compact strategy uses the probability distribution to simulate the population, so the algorithm only needs to maintain the parameters of the probability distribution, and the updating of the parameters replaces the updating of the population, which will greatly reduce the memory requirements of the algorithm. Therefore, if we want to reduce the burden of sensor node memory, using the compact strategy to improve will be the focus of future work.

## Figures and Tables

**Figure 1 sensors-23-04520-f001:**
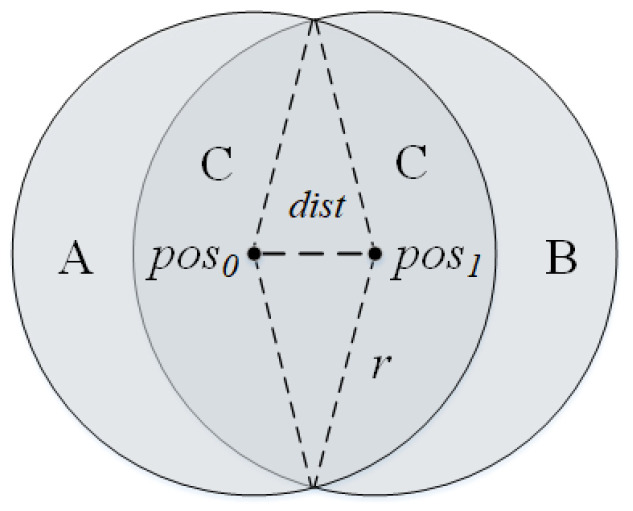
The anchor node is at $pos_{0}$ at time 0 and moves to $pos_{1}$ at time 1.

**Figure 2 sensors-23-04520-f002:**
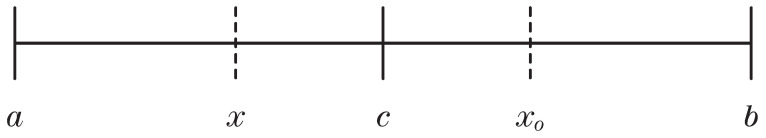
*c* is the midpoint of the interval [a,b], and $x_{o}$ is the opposite point of *x*.

**Figure 3 sensors-23-04520-f003:**
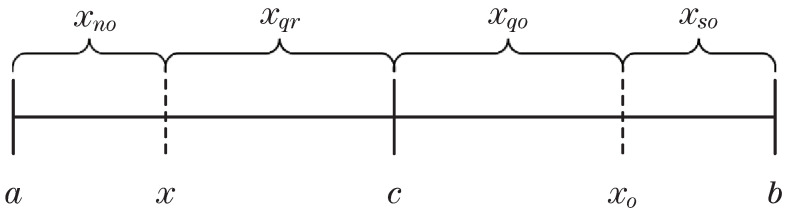
$x_{no}$, $x_{qo}$, $x_{so}$, and $x_{qr}$ are near opposition, quasi opposition, super opposition, and quasi reflective opposition of *x*, respectively.

**Figure 4 sensors-23-04520-f004:**
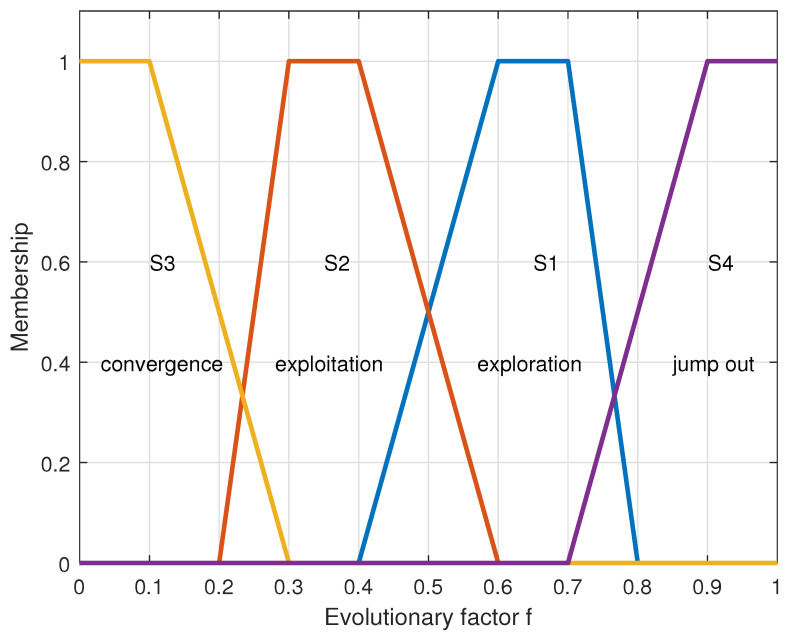
Fuzzy membership functions for the four evolutionary states.

**Figure 5 sensors-23-04520-f005:**
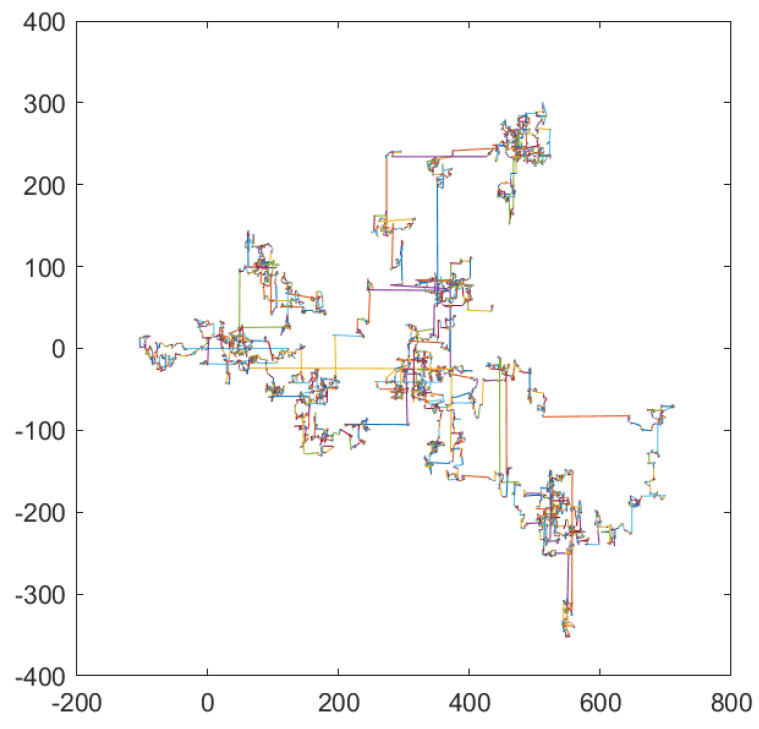
Step size of Levy flight.

**Figure 6 sensors-23-04520-f006:**
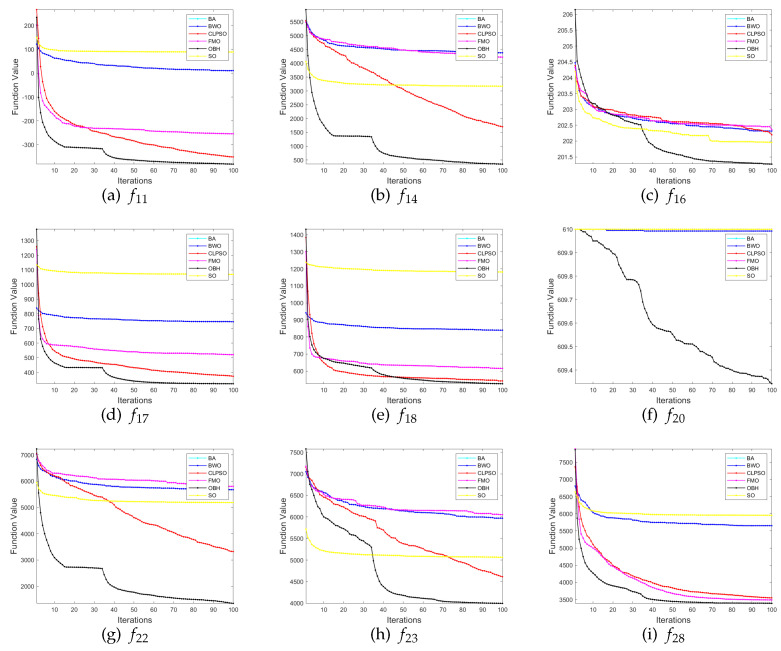
Comparison results between OBH and six common optimization algorithms.

**Figure 7 sensors-23-04520-f007:**
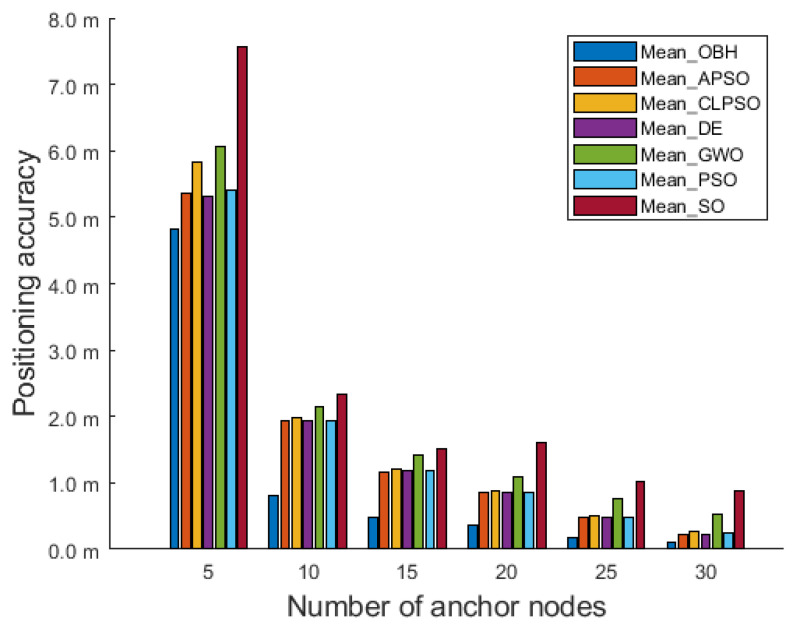
Comparison results in 200 m × 200 m × 200 m deployment area.

**Figure 8 sensors-23-04520-f008:**
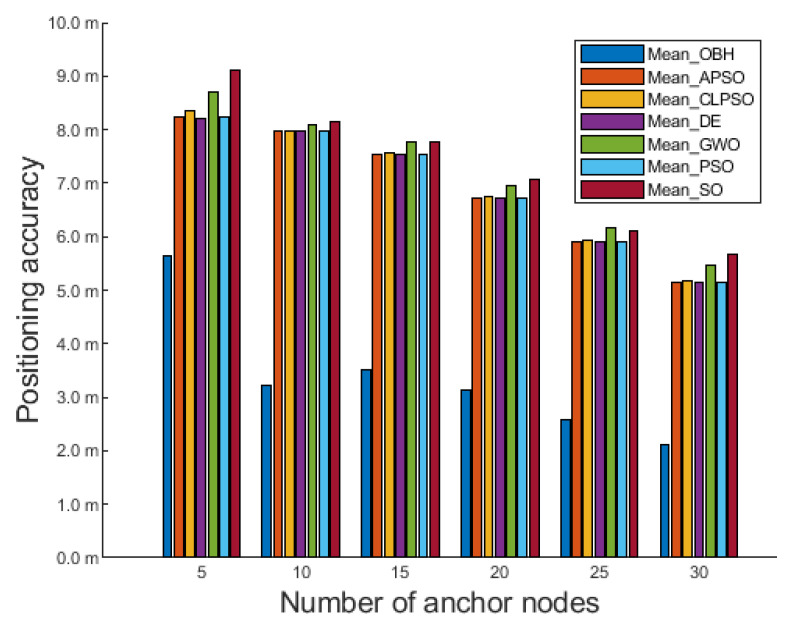
Comparison results in 400 m × 400 m × 400 m deployment area.

**Table 1 sensors-23-04520-t001:** Friedman’s on OBH and BH algorithms.

Algorithm	OBH		BH
Score	1.8214	(<)	3.5714

**Table 2 sensors-23-04520-t002:** Comparison of BH with the OBH.

Function	BH		OBH
$f_{1}$	5.67 × 10${}^{3}$	(<)	−1.40 × 10${}^{3}$
$f_{2}$	3.00 × 10${}^{7}$	(<)	1.49 × 10${}^{6}$
$f_{3}$	1.78 × 10${}^{14}$	(<)	7.80 × 10${}^{8}$
$f_{4}$	4.25 × 10${}^{4}$	(<)	5.51 × 10${}^{3}$
$f_{5}$	5.56 × 10${}^{2}$	(<)	−1.00 × 10${}^{3}$
$f_{6}$	4.43 × 10${}^{2}$	(<)	−8.77 × 10${}^{2}$
$f_{7}$	2.00 × 10${}^{4}$	(<)	−7.00 × 10${}^{2}$
$f_{8}$	−6.79 × 10${}^{2}$	(<)	−6.79 × 10${}^{2}$
$f_{9}$	−5.77 × 10${}^{2}$	(<)	−5.80 × 10${}^{2}$
$f_{10}$	1.63 × 10${}^{2}$	(<)	−4.98 × 10${}^{2}$
$f_{11}$	−1.06 × 10${}^{2}$	(<)	−3.84 × 10${}^{2}$
$f_{12}$	5.57 × 10${}^{0}$	(<)	−1.99 × 10${}^{2}$
$f_{13}$	1.32 × 10${}^{2}$	(<)	−4.46 × 10${}^{1}$
$f_{14}$	4.09 × 10${}^{3}$	(<)	3.67 × 10${}^{2}$
$f_{15}$	3.87 × 10${}^{3}$	(<)	2.74 × 10${}^{3}$
$f_{16}$	2.02 × 10${}^{2}$	(<)	2.01 × 10${}^{2}$
$f_{17}$	5.49 × 10${}^{2}$	(<)	3.22 × 10${}^{2}$
$f_{18}$	6.95 × 10${}^{2}$	(<)	5.23 × 10${}^{2}$
$f_{19}$	1.67 × 10${}^{3}$	(<)	5.02 × 10${}^{2}$
$f_{20}$	6.10 × 10${}^{2}$	(<)	6.09 × 10${}^{2}$
$f_{21}$	1.64 × 10${}^{3}$	(<)	1.01 × 10${}^{3}$
$f_{22}$	5.67 × 10${}^{3}$	(<)	1.37 × 10${}^{3}$
$f_{23}$	5.80 × 10${}^{3}$	(<)	4.11 × 10${}^{3}$
$f_{24}$	1.29 × 10${}^{3}$	(<)	1.26 × 10${}^{3}$
$f_{25}$	1.41 × 10${}^{3}$	(<)	1.36 × 10${}^{3}$
$f_{26}$	1.40 × 10${}^{3}$	(>)	1.51 × 10${}^{3}$
$f_{27}$	2.31 × 10${}^{3}$	(<)	2.14 × 10${}^{3}$
$f_{28}$	4.94 × 10${}^{3}$	(<)	3.48 × 10${}^{3}$
</=/>		1/0/27	

**Table 3 sensors-23-04520-t003:** Comparison of OBH with the BA, BWO, CLPSO, FMO, and SO.

Function	BA		BWO		CLPSO		FMO		SO		OBH
$f_{1}$	−1.40 × 10${}^{3}$	=	2.16 × 10${}^{4}$	<	−1.40 × 10${}^{3}$	=	−1.30 × 10${}^{3}$	=	−1.40 × 10${}^{3}$	<	−1.40 × 10${}^{3}$
$f_{2}$	1.96 × 10${}^{6}$	<	1.48 × 10${}^{8}$	<	7.58 × 10${}^{6}$	<	3.69 × 10${}^{6}$	<	4.57 × 10${}^{6}$	<	1.74 × 10${}^{6}$
$f_{3}$	1.80 × 10${}^{8}$	>	2.72 × 10${}^{16}$	<	8.70 × 10${}^{7}$	>	1.03 × 10${}^{9}$	<	3.14 × 10${}^{8}$	>	6.81 × 10${}^{8}$
$f_{4}$	6.06 × 10${}^{4}$	<	6.35 × 10${}^{4}$	<	3.95 × 10${}^{4}$	=	4.02 × 10${}^{4}$	=	2.69 × 10${}^{4}$	<	5.29 × 10${}^{3}$
$f_{5}$	−9.99 × 10${}^{2}$	=	4.73 × 10${}^{3}$	<	−1.00 × 10${}^{3}$	>	−1.00 × 10${}^{3}$	<	−9.98 × 10${}^{2}$	<	−1.00 × 10${}^{3}$
$f_{6}$	−8.60 × 10${}^{2}$	=	4.58 × 10${}^{3}$	<	−8.93 × 10${}^{2}$	>	−8.77 × 10${}^{2}$	=	−8.63 × 10${}^{2}$	<	−8.87 × 10${}^{2}$
$f_{7}$	1.30 × 10${}^{6}$	<	1.24 × 10${}^{5}$	<	−7.58 × 10${}^{2}$	>	−6.28 × 10${}^{2}$	<	−7.27 × 10${}^{2}$	>	−6.96 × 10${}^{2}$
$f_{8}$	−6.79 × 10${}^{2}$	<	−6.79 × 10${}^{2}$	<	−6.79 × 10${}^{2}$	<	−6.79 × 10${}^{2}$	=	−6.79 × 10${}^{2}$	<	−6.79 × 10${}^{2}$
$f_{9}$	−5.76 × 10${}^{2}$	<	−5.76 × 10${}^{2}$	<	−5.83 × 10${}^{2}$	>	−5.78 × 10${}^{2}$	<	−5.84 × 10${}^{2}$	>	−5.81 × 10${}^{2}$
$f_{10}$	−4.98 × 10${}^{2}$	<	2.06 × 10${}^{3}$	<	−4.91 × 10${}^{2}$	=	−4.77 × 10${}^{2}$	=	−4.75 × 10${}^{2}$	<	−4.99 × 10${}^{2}$
$f_{11}$	8.93 × 10${}^{1}$	<	1.11 × 10${}^{1}$	<	−3.52 × 10${}^{2}$	<	−2.54 × 10${}^{2}$	<	−3.56 × 10${}^{2}$	<	−3.82 × 10${}^{2}$
$f_{12}$	3.17 × 10${}^{2}$	<	1.72 × 10${}^{2}$	<	−1.78 × 10${}^{2}$	<	−1.39 × 10${}^{2}$	<	−2.26 × 10${}^{2}$	>	−1.88 × 10${}^{2}$
$f_{13}$	4.38 × 10${}^{2}$	<	2.77 × 10${}^{2}$	<	−7.00 × 10${}^{1}$	>	−3.17 × 10${}^{1}$	<	−6.20 × 10${}^{1}$	>	−5.58 × 10${}^{1}$
$f_{14}$	3.17 × 10${}^{3}$	<	4.38 × 10${}^{3}$	<	1.70 × 10${}^{3}$	<	4.23 × 10${}^{3}$	<	9.47 × 10${}^{2}$	<	3.50 × 10${}^{2}$
$f_{15}$	3.24 × 10${}^{3}$	=	4.44 × 10${}^{3}$	<	2.67 × 10${}^{3}$	>	4.24 × 10${}^{3}$	<	4.23 × 10${}^{3}$	<	2.95 × 10${}^{3}$
$f_{16}$	2.02 × 10${}^{2}$	<	2.02 × 10${}^{2}$	<	2.02 × 10${}^{2}$	<	2.02 × 10${}^{2}$	<	2.02 × 10${}^{2}$	<	2.01 × 10${}^{2}$
$f_{17}$	1.07 × 10${}^{3}$	<	7.46 × 10${}^{2}$	<	3.73 × 10${}^{2}$	<	5.21 × 10${}^{2}$	<	3.88 × 10${}^{2}$	<	3.22 × 10${}^{2}$
$f_{18}$	1.18 × 10${}^{3}$	<	8.40 × 10${}^{2}$	<	5.42 × 10${}^{2}$	=	6.16 × 10${}^{2}$	<	5.40 × 10${}^{2}$	<	5.25 × 10${}^{2}$
$f_{19}$	5.24 × 10${}^{2}$	<	6.10 × 10${}^{4}$	<	5.05 × 10${}^{2}$	<	5.06 × 10${}^{2}$	=	5.06 × 10${}^{2}$	=	5.02 × 10${}^{2}$
$f_{20}$	6.10 × 10${}^{2}$	<	6.10 × 10${}^{2}$	<	6.10 × 10${}^{2}$	<	6.10 × 10${}^{2}$	<	6.09 × 10${}^{2}$	>	6.09 × 10${}^{2}$
$f_{21}$	1.02 × 10${}^{3}$	>	2.11 × 10${}^{3}$	<	1.04 × 10${}^{3}$	<	1.07 × 10${}^{3}$	=	9.58 × 10${}^{2}$	>	1.02 × 10${}^{3}$
$f_{22}$	5.18 × 10${}^{3}$	<	5.68 × 10${}^{3}$	<	3.32 × 10${}^{3}$	<	5.80 × 10${}^{3}$	<	2.20 × 10${}^{3}$	<	1.35 × 10${}^{3}$
$f_{23}$	5.06 × 10${}^{3}$	<	5.97 × 10${}^{3}$	<	4.61 × 10${}^{3}$	<	6.05 × 10${}^{3}$	<	4.23 × 10${}^{3}$	<	4.00 × 10${}^{3}$
$f_{24}$	1.31 × 10${}^{3}$	<	1.30 × 10${}^{3}$	<	1.25 × 10${}^{3}$	>	1.26 × 10${}^{3}$	>	1.25 × 10${}^{3}$	>	1.26 × 10${}^{3}$
$f_{25}$	1.36 × 10${}^{3}$	=	1.38 × 10${}^{3}$	<	1.37 × 10${}^{3}$	<	1.37 × 10${}^{3}$	<	1.36 × 10${}^{3}$	<	1.36 × 10${}^{3}$
$f_{26}$	1.47 × 10${}^{3}$	<	1.44 × 10${}^{3}$	>	1.40 × 10${}^{3}$	>	1.44 × 10${}^{3}$	>	1.43 × 10${}^{3}$	>	1.46 × 10${}^{3}$
$f_{27}$	2.34 × 10${}^{3}$	<	2.39 × 10${}^{3}$	<	1.85 × 10${}^{3}$	>	2.11 × 10${}^{3}$	>	2.02 × 10${}^{3}$	>	2.17 × 10${}^{3}$
$f_{28}$	5.96 × 10${}^{3}$	<	5.65 × 10${}^{3}$	<	3.55 × 10${}^{3}$	<	3.49 × 10${}^{3}$	<	3.76 × 10${}^{3}$	<	3.39 × 10${}^{3}$
</=/>	2/5/21		1/0/27		10/4/14		3/7/18		10/1/17		−

**Table 4 sensors-23-04520-t004:** In a deployment area of 200 m × 200 m × 200 m, the average positioning accuracy of seven algorithms optimizing MCL.

Algorithm	OBH	APSO	CLPSO	DE	GWO	PSO	SO
Mean value	1.130	1.672	1.775	1.761	2.001	1.681	2.490

**Table 5 sensors-23-04520-t005:** In a deployment area of 400 m × 400 m × 400 m, the average positioning accuracy of seven algorithms optimizing MCL.

Algorithm	OBH	APSO	CLPSO	DE	GWO	PSO	SO
Mean value	3.366	6.922	6.961	6.919	7.191	6.923	7.312

## Data Availability

The data are contained within the article.

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
