# Peer review of "An Opposition-Based Learning Black Hole Algorithm for Localization of Mobile Sensor Network"

_sensors, 2023, doi:10.3390/s23094520_

Round 1

Reviewer 1 Report

1. Abstract is too brief. It should be elaborated to explain the problem statements and contributions of work. Results should be presented in qualitative manner.

2. Lines 34 to 47 - The classification of metaheuristic search algorithm mentioned in this paragraph is incorrect and outdated. According to recent publication (e.g., https://doi.org/10.1016/j.aej.2021.09.013), existing metaheuristic search algorithms are classified into four major categories, i.e., (a) Evolutionary algorithm, (b) Swarm intelligence algorithm, (c) Physics-based algorithm and (d) Human based algorithm. BH should belong to Physics-based instead of SI. Please rewrite this paragraph to ensure the information provided are updated.

3. Authors did not explain why metaheuristic search algorithms are suitable to solve the mobile node allocation problem as compared to those conventional methods such as particle filter algorithms and Monte Carlo localization. Please elaborate this part.

4. Problem statement and research gaps that motivated the current study are also not properly explained, particularly on the weakness of original Black Hole algorithm. Please elaborate.

5. Technical contributions and novelty of this manuscript are not mentioned at all. Please add. 

6. It is not clearly explained how BH or OBH can be used to solve the mobile node allocation problem. What is the relationship between OBH and Monte Carlo localization algorithm? Is OBH used to optimize certain parameters of Monte Carlo localization algorithm to solve the mobile node allocation problem? Or OBH can be used to solve the mobile node allocation problem by itself? Please clarify.

7. Figure 4 - How to determine these fuzzy membership functions. It is not clear what searching strategies are used in each stage (i.e., convergence, exploitation, exploration and jump out). Please summarize in pseudo-code or in table form.

8. Line 198 - Is it always true the states will changes from S1 => S2 => S3 => S4 => S1? How do the authors reach to this conclusion?

9. Figure 4 - What if the evolutionary factor has membership value between different states? For instance, if f  =0.5, then the membership values for Exploitation and Exploration are non-zero. What are the decision (e.g., search strategies to generate new solution) to be made for the OBH then?

10. Algorithm 3 and Section 3- Overall are not properly explained. For instance, it is not mentioned what are the solution encoding strategy used and what are the decision variables to be optimized? The linkage between the optimized algorithm and problem to be solved are not clearly explained. For instance, what are the fitness functions used to measure the quality of each solution?

11. Contents of proposed methodology are not well organized and confusing. Description of Algorithm 3 is not detailed enough because many important process in Section 3 are not mentioned. Please ensure all descriptions provided are captured. 

12. Section 4.1 - Authors should also compare the computational complexity between BH and OBH to justify the trade-off between performance gains and computational complexity incurred.

13. Section 5 is actually confusing between it is not mentioned what are the role of OBH and other optimization algorithm in Monte Carlo localization. Again, this is the main issue of this manuscript because the linkage between the optimized algorithm and problem to be solved are not clearly explained.

14. Quantitative results should be presented in Section 5. More in-depth discussion should be made to explain the performance differences between compared algorithms.

15. Figures 6 to 8 are too small. Needs to enlarge them for better readability. 

Reviewer 2 Report

Dear Authors

Thank you for this good works. It was pleasure to e to revise this paper.

Regards.

Paper Title: A opposition-based learning Black Hole Algorithm for localization of mobile sensor network

Paper ID:2332707

The paper is very well written, and contributes A opposition-based learning Black Hole Algorithm for localization of mobile sensor network. The following are the suggestions to improve the paper.

Update the literature with some recent year articles in the paper:

1.       Mohamed,A.A., Laith Abualigah, Alburaikan, A., and Khalifa, H.A. (2023). AOEHO: A new hybrid data replication method in fog computing for Iot application. Sensors (MDPI), 23(4):1289

Where are the highlights of your work and the novel aspects of this work?

Limitation of the study must be done.

Comparative study with respect to the existing methods must be done.

The fourth and fifth parts should be merged and well-explained.

The conclusion may be extended to future works.

As for the used algorithms in the second part of the former studies, it is supposed to be in a separate third part entitled BACKGROUND.

WSN needs to be added to the keywords in addition to the important words.

In the third part clarification of the frame is needed and explained then the suggested algorithm should be well-explained.

Reviewer 3 Report

For the present version of the paper it is hard to evaluate the contribution of the authors. English correction service is highly recommended. The explanation of motivation needs to be improved. In WSN, localization can be done in various ways (GPS, beacon signals, triangulation, etc.) Authors need to point specific applications for their proposals, where this architecture (anchor nodes) and the proposed localization are applied.

System model and proposed ideas are not well introduced. What is the relationship between fitness and position? In what units are they measured? How is the problem of localization related to the process of absorption of a star by a black hole? Why is it necessary to maintain a constant population size and randomly generate a new star?

The variable p is used for both position and prediction probability. Does p mean the probability of predicting the true location? What deviations are allowed? How to calculate the mentioned probabilities? What exactly does "observation of anchor nodes at the time t" include? Are L_t and l_t the same? Figure 2 rather confuses the situation. It contradicts formulas (8) and (9). Probably, x0=rand(c, x0). Also, it needs to properly introduce x_n0. What does x mean? How is this entity related to the problem at hand?

A detailed description of the variable D and index k in formula (11) is required. Since the “optimal value”, “optimal position” is used, it is necessary to give the statement of the optimization problem corresponding to them.

Round 2

Reviewer 1 Report

Authors have addressed most of the comments given in earlier review process appropiately. No further comments from reviewers.